# The Association between Vitamin D and Gut Microbiota: A Systematic Review of Human Studies

**DOI:** 10.3390/nu13103378

**Published:** 2021-09-26

**Authors:** Federica Bellerba, Valeria Muzio, Patrizia Gnagnarella, Federica Facciotti, Susanna Chiocca, Paolo Bossi, Diego Cortinovis, Ferdinando Chiaradonna, Davide Serrano, Sara Raimondi, Barbara Zerbato, Roberta Palorini, Stefania Canova, Aurora Gaeta, Sara Gandini

**Affiliations:** 1Department of Experimental Oncology, IEO European Institute of Oncology IRCCS, 20141 Milan, Italy; federica.bellerba@ieo.it (F.B.); federica.facciotti@ieo.it (F.F.); susanna.chiocca@ieo.it (S.C.); sara.raimondi@ieo.it (S.R.); Aurora.Gaeta@ieo.it (A.G.); sara.gandini@ieo.it (S.G.); 2Division of Epidemiology and Biostatistics, IEO European Institute of Oncology IRCCS, 20141 Milan, Italy; valeria.muzio@studenti.unimi.it; 3Medical Oncology, Department of Medical and Surgical Specialties, Radiological Sciences and Public Health University of Brescia, ASST-Spedali Civili, 25121 Brescia, Italy; paolo.bossi@unibs.it; 4SC Oncologia Medica, Asst H S Gerardo Monza, 20900 Monza, Italy; d.cortinovis@asst-monza.it (D.C.); s.canova@asst-monza.it (S.C.); 5Department of Biotechnology and Biosciences, University of Milano-Bicocca, 20126 Milan, Italy; ferdinando.chiaradonna@unimib.it (F.C.); barbara.zerbato@unimib.it (B.Z.); roberta.palorini@unimib.it (R.P.); 6Department of Cancer Prevention and Genetics, IEO European Institute of Oncology IRCCS, 20141 Milan, Italy; davide.serrano@ieo.it

**Keywords:** microbiome, vitamin D, supplementation, 25(OH)D, prevention

## Abstract

Recent evidence has shown a number of extra-skeletal functions of Vitamin D (VD), primarily involving the immune system. One of these functions is mediated by the modulation of gut microbiota, whose alterations are linked to many diseases. Our purpose is to contribute to the understanding of existing evidence on the association between VD and gastrointestinal microbiota alterations. A systematic review of studies with human subjects has been conducted up to January 2021. We included publications reporting the association between gut microbiota and VD, including VD supplementation, dietary VD intake and/or level of 25(OH)D. We identified 25 studies: 14 were interventional and 11, observational. VD supplementation was found to be associated with a significant change in microbiome composition, in particular of *Firmicutes*, *Actinobacteria* and *Bacteroidetes* phyla. Furthermore, *Firmicutes* were found to be correlated with serum VD. Concerning alpha and beta diversity, a high nutritional intake of VD seems to induce a shift in bacterial composition and/or affects the species’ richness. *Veillonellaceae* and *Oscillospiraceae* families, in the *Firmicutes* phylum, more frequently decreased with both increasing levels of 25(OH)D and vitamin D supplementation. We found evidence of an association, even though the studies are substantially heterogeneous and have some limitations, resulting sometimes in conflicting results. To further understand the role of VD on the modulation of the gastrointestinal microbiota, future research should be geared toward well-designed animal-based studies or larger randomized controlled trials (RCTs).

## 1. Introduction

Vitamin D (VD) is mainly known for its role in skeletal homeostasis, through the calcium and phosphate absorption at the intestinal level mediated by its active form, the 1,25-dihydroxyvitamin D (1,25(OH)2D). Its presence is necessary for bone growth and bone health, as evidenced by the onset of rickets in infants and osteoporosis in elderly people with VD deficiencies. VD is present in two forms: VD3 and VD2, which differ for a double bond between C22 and C23, and a methyl group at the C24; this affects the bioavailability of the two forms, with the VD3 that is adsorbed more easily at the intestinal level. The dietary intake needs of VD is mostly satisfied by the consumption of fatty fish, which is the richest source, followed by egg yolk, liver, meat and fortified dairy products [1,2]. Nonetheless, the greatest amount of VD is synthesized at skin level from 7-dihydrocholesterol, after UVB exposure. Two different hydroxylations, made at the liver and kidneys’ level, lead to the active form of VD, whose synthesis is regulated by the activity of the enzyme CYP27B1. The presence of the enzyme CYP27B1 and of the vitamin D receptor (VDR), an ancient nuclear receptor, also found in other cells besides those of the kidneys, led us to consider different VD functions that are not exclusively related to bone metabolism [1]. In particular, the relationship between VD, the immune system, cardiometabolic disorders, cancer risk, and overall mortality highlights the potential role of vitamin D for the prevention and treatment of several chronic diseases in pre-clinical, epidemiological and clinical studies [3,4,5,6]. 

However, the promising results from the growing literature on the associations between VD and extra-skeletal chronic diseases is not matched by the results obtained in intervention studies. To prove a causal relationship and recommend the use of VD, it is important to understand VD extra-skeletal functions that primarily involve the immune system. One of these functions is mediated by the modulation of gut microbiota, whose alterations are linked to many diseases, like cardiovascular disease, diabetes and cancer.

VDRs mediate the biological actions of the active form of VD3 and it is extensively expressed in the gut, playing an important role in immune regulation, proliferation and intestinal homeostasis [7]. However, little is known about the direct effects of vitamin D on bacteria. VD inhibits the growth of specific mycobacterial species in vitro [8], implicating that the antimicrobial effects of vitamin D would be consistent with known immunoregulatory properties. Jahani et al. [9] show that in mice exposed to high levels of vitamin D3 during pregnancy and lactation, a lower level of vitamin D resulted in reduced vitamin D receptors and increased expression of pro-inflammatory genes in the colon at 3 months, lower numbers of colonic *Bacteroides/Prevotella* at postnatal day 21, and higher serum LPS concentration at adulthood. These results are confirmed by other rodent studies demonstrating that vitamin D deficiency through dietary restriction, lack of CYP27B1, or lack of VDRs promote increases in the *Bacteriodetes* [10,11,12,13] and *Proteobacteria* phyla [10,11,13]. Furthermore, a genome-wide association study (GWAS), carried out in a combined cohort of 2029 individuals, identified two VDR polymorphisms as significant contributors to microbiota variation [14]. The human VDR polymorphisms consistently influenced the genus *Parabacterioides* (phylum: *Bacterioidetes*), and the subsequent evaluation of VDR−/− mice showed a corresponding increased abundance of *Parabacteroides* compared to WT mice [14].

The gastrointestinal tract is inhabited by a large number of microorganisms, whose number was estimated to exceed 1014 [15]. “Microbiome” is a term that indicates the collection of the genome of these microorganisms, which is more than 100 times the number of the human genome [16]. Instead, the term microbiota indicates the multitude of microbial communities, including not only bacteria but also yeasts and viruses [17]. New technologies have led to the identification of these communities and to taxonomic classification. Almost 12 phyla were recognized, of which the most predominant are *Firmicutes*, *Bacteroidetes*, *Actinobacteria*, and *Proteobacteria*. The composition of the microbiota varies according to different gastrointestinal tracts, with the largest amounts of microbial communities found in the colon [18].

Its development begins during the gestational stage, and it is enhanced in the first years of life to progressively change in adulthood and in older age. In addition to age, many factors influence gut microbiota composition, like environment, exercise, lifestyle, diet, and antibiotics’ and general prescription use [19].

The microbiota is involved in maintaining the integrity of the intestinal mucosal barrier, providing protection against pathogens, supplying vitamins and metabolites, and shaping and regulating intestinal endocrine functions, neurologic signaling and the immune response. The latter function seems to be the key point that relates the condition of dysbiosis to several diseases like cancer, diabetes, and cardiovascular or autoimmune diseases [20]. Accordingly, the action of VD in modulating the gut’s microbial composition plays a relevant role in maintaining immune system function and, consequently, human health [7].

Since it is necessary to expand our knowledge on the effects that VD deficiency or supplementation may have on gut microbiota, this review aims to additionally contribute to the current evidence [21].

## 2. Materials and Methods

The search strategy was set according to PRISMA guidelines and was conducted up until January 2021. We consulted the following databases: Pubmed, EMBASE, CINAHL, and Cochrane, including all peer-reviewed articles published in English. We searched the keywords “vitamin D”, “vitamin D3”, “cholecalciferol” and “25 Hydroxyvitamin D” in combination with “microbiota”, “gut microbiota”, “microbiome” or “dysbiosis”.

The eligibility criteria were based on the PICOS framework [22]. Regarding the population, we selected all the human studies and considered participants of all ages, both healthy and non-healthy. We included interventional and observational studies, considering supplementation with VD, the assessment of dietary VD intake, and the measurement of serum 25(OH)D levels. In interventional studies, in single-arm trials, the results after VD supplementation were compared to baseline, whereas in two-arm studies the supplemented groups were compared to either a placebo or non-supplemented group.

The outcome was the analysis of microbiota composition by pyrosequencing the 16S rRNA gene, including the assessment of alpha-diversity, beta-diversity, species richness and the prevalence of bacterial taxa.

### 2.1. Data Extraction

One researcher extracted data (VM) using a data abstraction form, including information about the country in question, study design, population, outcomes, health status, the dose of VD supplementation or measure of dietary VD intake, serum 25(OH)D levels, and analysis of microbiota composition, in accordance with PICO’s criteria. The data were checked by a second researcher (F.B.). All questions and divergences regarding the data were discussed during a meeting.

### 2.2. Methods

We reviewed the studies separately by means of the comparison, health status and type of samples:Aim of the analysis:
-Effect of vitamin D supplementation;-Association with VD serum values (25OHD or 1,25OHD) or Vitamin D intake;Health status:-Healthy subjects;-Subjects with possible dysbiosis, pregnancies, obesity, diabetes;Types of samples for the microbiome:-Stool;-Biopsies.

A phylogenetic classification was provided for each significant taxa in summary tables, and the percentage of taxa in each phylum over the total number of significant taxa was calculated and graphically represented in frequency plots and mirror bar chats, according to whether the taxa abundances decreased or increased in relation to vitamin D, shown separately for each study group (supplementation studies or studies evaluating vitamin D serum levels or dietary intake). Where useful, similar plots were provided at the family level for a given phylum.

In addition, the phylogenetic trees of taxa were presented only for healthy subjects and microbiome evaluation in stool samples.

For the supplementation group of studies, the phylogenetic trees show taxa that significantly changed after VD supplementation, stratified according to whether the taxa abundances increased or decreased after the supplementation occurred. For the association between microbiota and VD serum values, the phylogenetic trees included all the taxa that were significantly associated with VD levels, either positively or negatively.

For each taxonomic level, the number of studies in which the given level belonged to at least one of the significant taxa was provided in the plots.

## 3. Results

Appendix A represents the PRISMA flowchart of the selection process for the studies included in the review. After eliminating the duplicates, we found 955 publications, of which 30 were assessed for eligibility. After the revisions of the selected full-text articles, a further 5 were excluded because they lacked adequate data on VD supplementation or showed the analysis of vaginal microbiota as an outcome [14,23,24,25,26]. As a result, this review covers 25 studies.

### 3.1. Qualitative Synthesis

#### 3.1.1. Study Designs

As described in Table 1, we found 14 interventional studies, 7 of which were randomized controlled trials (RCTs) [27,28,29,30,31,32,33]. The other 11 were observational studies, of which 4 were cohorts [34,35,36,37], 6 were cross-sectional [38,39,40,41,42,43] and 1 was a case-control study [44].

Most of the studies (17) were conducted in the United States and in Europe, 4 were in East Asia and in the Middle East, 2 in Canada, and the remaining 2 in Africa and Brazil.

The populations selected were predominantly healthy and were mostly composed of volunteers and students. Five studies were conducted on cohorts of pregnant women [29,34,35,36,37]. The other studies enrolled participants with ulcerative colitis and Crohn’s disease [38,44,45,46], followed by multiple sclerosis [47], cystic fibrosis [30], HIV [31], and prediabetes [27]. Only one study was conducted on an overweight/obese population [32].

In all the interventional studies, the length of the VD supplementation period was on average of 14–15 weeks. The doses ranged from a minimum of 400 UI/day to a maximum of 10,000 UI/day.

Eight studies enrolled participants with serum levels of 25(OH)D less than 30 ng/mL (VD deficiency) [27,28,30,32,45,46,47,48].

The analysis of microbiota was performed on stool samples, except for four studies that also examined gastrointestinal tissue biopsies [31,38,44,49]. DNA extraction and amplification and 16s ribosomal RNA sequencing presented great heterogeneity, mostly because of the different hypervariable regions analyzed. Moreover, only two studies conducted metagenomic sequencing by shotgun [43,44]. Alpha and beta diversity were evaluated with different metrics; the most used metrics were the Shannon index for alpha diversity and the weighted Unifrac distance for beta diversity.

Table 2 depicts the main significant changes in 25(OH)D levels and in alpha and beta diversity after VD supplementation and Table 3 reports the association between dietary VD intakes or serum 25(OH)D levels and alpha and beta diversity in the selected studies. VD supplementation determined a general increase of measured serum levels of 25(OH)D, as shown in the interventional studies [27,28,30,32,45,46,47,48,49,50].

Regarding alpha diversity, only 7 studies have found an association with VD, but the results were in some cases discordant [32,35,36,42,46,48,51] (Table 2 and Table 3). Indeed, some of the interventional studies showed a decrease in richness after VD supplementation [32,46]. Conversely, in the study by Bosman et al. [51], the non-supplemented group showed significantly lower diversity and richness before UVB exposure than the supplemented group. Similar results were also described by Singh et al. [48], where a significant increase in alpha diversity after VD supplementation was registered only in observed Operational Taxonomic Units (OTUs) and Chao1 indices, but not in the Shannon Index analysis. In cohorts of pregnant women, either maternal serum 25(OH)D or dietary VD intake were significantly and inversely associated with infant richness and diversity [35,36] (Table 3). Significant changes in beta diversity were found after VD supplementation in biopsies of the upper gastrointestinal tract, but not in fecal samples or in lower gastrointestinal biopsies [49] (Table 2).

Some evidence showed significant shifts in bacterial community composition, related to both VD supplementation and serum 25(OH)D levels [27,30,31,35,42,48] (Table 2 and Table 3).

#### 3.1.2. Distribution of Taxa at Phylum Level

The most recurrent phyla increasing after vitamin D supplementation were *Firmicutes* and *Bacteroidetes*, followed by *Proteobacteria* and *Actinobacteria* (Figure 1). A similar distribution was observed in the group of studies evaluating microbiome modulation with vitamin D serum levels or dietary intake changes. *Firmicutes* was the most recurrent decreasing and increasing phylum. For this reason, a further inspection at the family level was carried out. As shown in Figure 2, a large variety of families in *Firmicutes* was observed after vitamin D supplementation. The taxa of *Veillonellaceae* and *Oscillospiraceae* families more frequently decreased with increasing levels of 25OHD or with vitamin D supplementation. Since we found both an increase and decrease in the *Lachnospiraceae* family, we further investigated the differences at the genus level. The genus *Blautia* was mostly observed to be decreasing with both vitamin D supplementation and increasing 25(OH)D values. On the other hand, Thomas et al. [42] found a significant positive correlation between the species *Blautia obeum*, a member of the genus *Blautia,* and the active form of vitamin D, 1,25(OH)D. The genus *Roseburia* was also inversely correlated with vitamin D levels, with the exception of the positive correlation found by Singh et al. [48] in the group of non-responders to supplementation (25(OH)D: < 20 ng/mL at follow-up). Conversely, *Funicanibacter*, *Lachnospira* and *Lachnobacterium* were significantly more abundant in supplemented subjects, with the genus *Funicanibacter* positively correlated with vitamin D dietary intake in the Crohn’s disease cohort reported by Weng et al. [44].

Conflicting results were found for the genus *Coprococcus*, which was significantly more abundant in the supplemented cohort of untreated Multiple Sclerosis women patients [47] and in the subjects with a high response to the supplementation (25(OH)D: > 75 ng/mL vs. < 50 ng/mL) [32]. It correlated positively with 1,25(OH)D and the activation ratio in Thomas et al. [42], but inversely correlated with 25(OH)D levels in the healthy cohort by Luthold et al. [40]. Similarly, *Ruminococcus gnavus*, in the *Mediterraneibacter* genus, was positively correlated with prenatal maternal 25(OH)D but inversely correlated with infant cord 25(OH)D levels [35].

#### 3.1.3. Analysis of Phylogenetic Trees of Studies in Healthy Subjects

In supplementation studies, the most recurrent phyla that either increased or decreased significantly after VD supplementation were *Firmicutes, Actinobacteria* and *Bacteroidetes* (Appendix A). In *Firmicutes* phylum, several core genera from the *Lachnospiraceae* family, like *Lachnospira, Fusicatenibacter* and *Lachonacterium,* increased after supplementation, whereas two studies reported a decrease in the *Faecalibacterium* genus from the *Oscillospiraceae* family [28,48]. Moreover, several genera from the *Lactobacillales* order, such as *Lactococcus* and *Lactobacillus*, decreased after supplementation, except for *Enterococcus*, which increased in the female cohort of adolescents in the study by Tabatabaeizadeh et al. [50].

Increasing abundances were also found in *Actinobacteria* phylum, in particular in the *Bifidobacterium* genus, in other genera from *Bacteroidetes*, such as *Bacteroides*, *Parabacteroides* and *Alistipes*, and in *Akkermansia* from the *Verrucomicrobia* phylum.

In the non-supplementation group of studies (Appendix A), the associations between microbiota and VD concentrations in the serum were also investigated. In Luthold et al., *Veillonella* (*Firmicutes* phylum) and *Haemophilus* (*Proteobacteria* phylum) were found to be significantly more abundant in the lowest compared to the highest tertile of both VD intake and serum 25(OH)D levels [40]. Despite their beneficial effect, also *Coprococcus* (*Firmicutes* phylum) and *Bifidobacterium* (*Actinobacteria*) genera were inversely associated with 25(OH)D levels, even after adjustment for confounders. On the other hand, the *Megasphera* genus from the *Negativicutes* order (*Firmicutes* phylum) was significantly more abundant in the highest tertile of 25(OH)D levels compared to the lowest [40]. In the community-dwelling older-men cohort studied by Thomas et al., they analyzed the association of microbiota not only with 25(OH)D concentrations but also the active form of VD, 1,25(OH)2D, and the ratio of activation (1,25(OH)2D/25(OH)D) [42]. While they found no significant association between 25(OH)D and any microbiota measure or specific OTUs, they observed a significant positive association between *Coprococcus catus* and *Blautia Obeum* species (*Firmicutes* phylum; *Clostridia* class) and 1,25(OH)2D, and between the *Eubacteriales* order, *Ruminococcaceae*, *Lachnospiraceae*, *Victivallaceae* families, *Coprococcus* and *Mogibacterium* genera, and the ratio of active VD. Conversely, *Blautia* and *Oscillospira*, belonging to the *Firmicutes* phylum and *Clostridia* class, were significantly and negatively associated with both 1,25(OH)2D levels and the VD active ratio, whereas the *Anaerotruncus* genus was associated only with 1,25(OH)2D.

Appendix A show a phylogenetic reconstruction of taxa that significantly decreased or increased after vitamin D supplementation, according to health status and type of microbiome samples. Moreover, Appendix A illustrate the phylogenetic reconstruction of taxa that were significantly positively or negatively associated with either vitamin D serum concentrations or vitamin D intake, by health status and type of microbiome samples.

One of the five RCTs demonstrated that VD supplementation had a dose-related effect on microbiota composition, which was highlighted by significant differences in the increased abundance of *Bacteroides* and *Parabacteroides* in the groups of interventions [28].

As is consistent with this, in overweight or obese patients, a first loading dose of cholecalciferol (100,000 UI) followed by 4000 UI/day was significantly associated with a higher abundance of the genus *Lachnospira* and lower abundance of the genus *Blautia* [32].

Two studies differed in their encountered significance. In one study, the changes after supplementation occurred only in the superior GI tract and were detected in biopsies but not in fecal samples [49]. Instead, Bosman et al. showed significant results only after the exposure to narrow-band ultraviolet B (NB-UVB) light, especially for the VD-deficient group [51].

Concerning the results of one of the selected cohorts, which mostly referred to mother-child couples, prenatal higher doses of VD were related to significant changes in infant microbiota composition, with a decreased abundance of *Bilophila* and *Lacnospiraceae* [30]. Infant VD supplementation did not show a significant effect, except for the lower abundance of the genus *Megamonas* in gut microbiota, as reported in the study by Drall et al. [30]. Talsness et al. found that *Bifidobacterium* abundance was inversely related to higher levels of maternal serum 25(OH)D [37].

More significant changes in the microbial taxa occurred in inflammatory bowel disease patients. One interventional non-randomized trial [46] showed a significant increase of the phylum *Firmicutes* after VD supplementation in patients with Crohn’s disease, whereas no change was observed in the healthy control group. A positive correlation between *Firmicutes* and dietary vitamin D intakes in Crohn’s disease patients was also assessed in a case-control study [44]. Moreover, the seasonal variability in serum 25(OH)D was found to be positively correlated with *Proteobacteria* in patients with Crohn’s disease in one observational study [38], but no significant association was found in the ulcerative colitis group. Conversely, two studies on ulcerative colitis patients found significantly increasing levels of *Enterobacteriaceae* after VD supplementation and a significant positive correlation between the *Desulfovibrio* genus and levels of dietary VD intakes, and both taxa belong to the *Proteobacteria* phylum [44,45].

Among the selected studies, three did not highlight any associations between VD and gut microbiota composition. One study was an RCT [31], conducted on a group of HIV patients, in which the supplementation with a dose of 5000 UI/day for 16 weeks did not modulate the microbial gut composition. In one cross-sectional study [41], the analysis of dietary VD intake in a population of young Japanese women did not show any significant correlation with changes in fecal microbiota. Similarly, a large UK twins cohort study [39], characterized by the self-reported use of VD supplements, was also not significantly associated with a change in gut microbiota.

## 4. Discussion

Our findings suggest the presence of an association between VD and microbiota composition, even though the studies are substantially heterogeneous. The most relevant changes affect the main phyla of the intestinal microbial community, *Firmicutes*, *Actinobacteria* and *Bacteroidetes*, with either a decrease or an increase in relative abundance. Regarding alpha and beta diversity, a high dietary intake of VD seems to induce a shift in bacterial community composition in some studies, and also affects the species richness.

These findings are supported by recent evidence that presumes an indirect effect of VD on gut microbiota composition [7,52,53]. The presence of VDRs at the gastrointestinal level could be one of the mechanisms by which VD can influence microbial composition, especially through the regulation of the immune response [54] as reported in Wang’s GWAS, in which VDR polymorphisms were related to the abundance of the genus *Parabacteroides* [14]. VDRs also mediate the production of antimicrobial peptides, like cathelicidin and defensin [55,56] similarly the active form of VD induces their production at macrophage level [57]. These antimicrobial peptides play a relevant role in the maintenance of microbial homeostasis. Furthermore, existing evidence hypothesized the effect of VD and VDRs in the regulation and preservation of gut integrity and microbiota functions through the suppression of beta-catenin, which promotes cell proliferation, and through the enhancement of the expression of tight junctions’ proteins, like E-cadherin, Occludin and ZO-1 [58]. However, our review presents some limitations.

Firstly, we found only 7 RCTs [27,28,29,30,31,32,33], and the reliability of these studies is affected by some aspects, such as the small size and the different characteristics of the enrolled populations, which often led to conflicting results. Regarding sample size, only one RCT had a large population, including more than 500 participants with different health conditions from the COPSAC10 cohort [29]. The small sample size is a common limitation that also applies to the observational studies, as only four studies enrolled more than 500 participants [34,37,39,42].

Secondly, comparing the different studies on healthy subjects and the association between VD level and microbiome composition, we should also consider that the results of the community-dwelling older-men cohort by Thomas et al. may not be comparable with the other studies, since the microbiome in older men may be different from young healthy subjects [42]. We know that infection-related morbidities are the leading cause of death in the elderly, and respiratory pathogens, severe acute respiratory syndrome (SARS), and its sister coronaviruses frequently cause enteric symptoms. Similarly, other classically non-enteric viruses, such as HIV and influenza, also have enteric effects that are crucial in their pathogeneses. These effects can be due to direct infection of the gut mucosa but can also be because of decreased antibacterial defenses, increased mucosal permeability, bacterial translocation, and the systemic leak of endotoxins. The microbiota could confer protection against viral infection by priming the immune response as a way to avoid infection, with some bacterial species being required to increase the antiviral response. On the other hand, it could also help to promote the viral evasion of certain viruses by direct and indirect mechanisms, with the presence of the microbiota increasing infection and viruses using lipopolysaccharides (LPS) and surface polysaccharides from bacteria to trigger immunosuppressive pathways. The integrity of the commensal microbiota can be disturbed by invading viruses, causing dysbiosis in the host and further influencing virus infectivity.

The analysis of gut microbiota raises other questions about the possibility of extrapolating relevant results. Regarding the studies’ design, it is known that an RCT is less subject to biases, but all the studies included in this review are characterized by a very small sample size, which can hardly represent the general population’s outcomes. However, in observational studies with a larger sample size, confounding factors were not taken into consideration, such as environmental factors, lifestyle, dietetic patterns, or different times of sampling that could alter the microbiota composition [59]. Reverse causation should also be considered in the interpretation of studies evaluating the association with 25OHD since the microbiome could also influence health status and, therefore, serum biomarkers. At the sequencing levels, heterogeneity between studies is enhanced according to the chosen method. In almost all the studies, a gene amplicon sequencing of 16s RNA was used, but the variable regions under consideration ranged widely. Indeed, this could be a relevant factor because every single region has a different accuracy in identifying microbial profiles [60]. In contrast, a genome-wide shotgun sequencing approach ensures a better taxonomic analysis and genomic information but this method was used in only two of the selected studies for this review, and only on subgroups of patients [43,44].

Singh [48] was able to predict functional profiling of the gut microbial communities pre- and post- vitamin D supplementation using PICRUST analysis. The authors found marked differences between predicted patterns of functional genes pre- and post- vitamin D supplementation, and significant differences in the genes related to host-symbiont metabolic pathways, including folate biosynthesis and glycine, serine and threonine metabolism. The increase in the abundance of *Bifidobacterium*, which is able to produce folate, may explain the predicted increase of folate biosynthesis. Singh also predicted an increase in genes related to the pathways involved in lipid metabolism and fatty acid biosynthesis, which have a role in the absorption of vitamin D (fat-soluble) in the intestinal lumen [48].

According to these considerations, to better understand and evaluate the association between VD and microbiota, it is necessary to design more accurate RCTs, which should include a larger population. To avoid biases and to achieve a better interpretation of the results, the choice of sample size should be based on rigorous statistical principles. In addition, the enrollment of participants and the procedures of sampling and analysis of the microbiome should also be established according to shared best practices [59]. Mouse model studies could also help to better understand the functional mechanisms, for example, by using, in certain pathological contexts, VD supplementation in concomitance with specific systems to evaluate the function and composition of the microbiota (e.g., using GF mice, or treating with selective antibiotics, or using stool transplantation from patients treated or not treated with VD). For example, Bora et al. investigated the effect of microbiota on vitamin D metabolism in germ-free mice and found several factors responsible for the modulation of this relationship, such as FGF23, which decreased vitamin D levels, highlighting the importance of investigating the mechanisms of this process [61].

Finally, diet, microbiome, vitamin D, markers of inflammation and adipokines are strongly connected in a complex network, and the unbalance of one or more factors may contribute to cancer incidence and prognosis. We observed a significant reduction in colorectal risk when comparing the highest versus the lowest level of serum 25-OHD, with a significant dose–response effect [62] and a significant association with VDR polymorphisms [63]. More recently, we evaluated the role of the microbiome and diet in colorectal etiology, taking into account vitamin D and other biomarkers. Results suggest that the microbiome mediates the effect of diet on CRC risk and that VD, markers of inflammation, and adipokines are other factors to consider in order to achieve a better knowledge of the whole carcinogenic process [64].

## 5. Conclusions

This review leads to a better understanding of the association between VD and gut microbiota and identifies consistent results through the existing published literature. Even though in the last few years, several studies have documented this relationship, we did not carry out a formal meta-analysis because of between-study heterogeneity, mainly due to different health conditions and the types of microbiome samples.

A qualitative review outlines the fact that vitamin D supplementation prompts a significant change in the abundance of *Firmicutes*, *Actinobacteria* and *Bacteroidetes*. Functional profiling of the gut microbial communities identified metabolic pathways that are involved in lipid metabolism and fatty acid biosynthesis, which have a role in the absorption of vitamin D (fat-soluble) in the intestinal lumen.

Further research should better highlight the mechanisms that are the basis of the gut microbiota compositional change according to VD supplementation or status. This could be achieved with well-designed animal models and RCTs, guaranteeing consistency in gut microbiota analysis and in VD supplementation or assessment.

## Figures and Tables

**Figure 1 nutrients-13-03378-f001:**
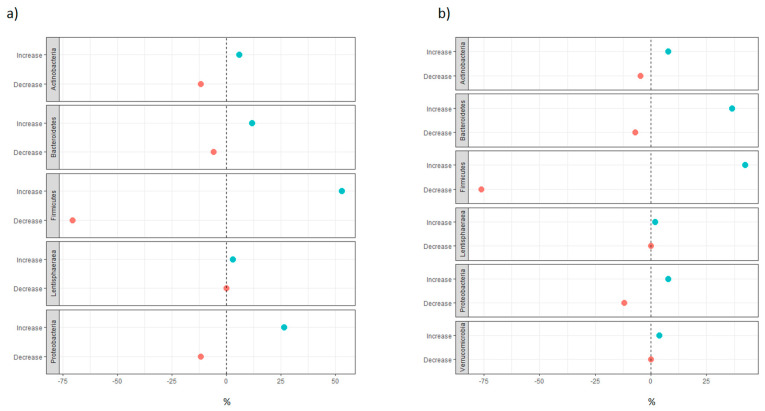
(**a**) For each phylum, the blue dot indicates the number of significant taxa in the phylum that increased with increasing levels of VD serum levels or dietary intake, over the total number of significant taxa that increased with increasing levels of VD serum levels or dietary intake in the non-supplementation group of studies (expressed in percentages); the red dot indicates the number of significant taxa in the phylum that decreased with increasing levels of VD serum levels or dietary intake, over the total number of significant taxa that decreased with increasing levels of VD serum levels or dietary intake in the non-supplementation group of studies (expressed in percentages). (**b**) For each phylum, the blue dot indicates the number of significant taxa in the phylum taxa that increased after VD supplementation, over the total number of significant taxa that increased after VD supplementation in the supplementation group of studies (expressed in percentages); the red dot indicates the number of significant taxa in the phylum that decreased after VD supplementation, over the total number of significant taxa that decreased after VD supplementation in the supplementation group of studies (expressed in percentages).

**Figure 2 nutrients-13-03378-f002:**
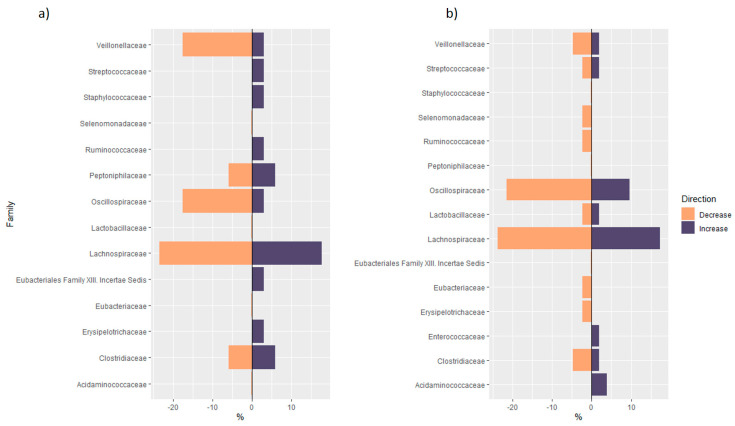
(**a**) mirror bar chart of the number of significant taxa in each family of Firmicutes phylum over the total number of significant taxa in each family of Firmicutes phylum found in the non-supplementation group of studies: for each family, the violet bar indicates the number of significant taxa in the family that increased with increasing levels of VD serum levels or dietary intake, over the total number of significant taxa in Firmicutes that increased with increasing levels of VD serum levels or dietary intake (expressed in percentages); the orange bar indicates the number of significant taxa in the family that decreased with increasing levels of VD serum levels or dietary intake, over the total number of significant taxa in Firmicutes that decreased with increasing levels of VD serum levels or dietary intake (expressed in percentages); (**b**) Mirror bar chart of the number of significant taxa in each family of Firmicutes phylum over the total number of significant taxa of Firmicutes phylum found in the supplementation group of studies: for each family, the violet bar indicates the number of significant taxa in the family that increased after VD supplementation, over the total number of significant taxa in Firmicutes that increased after VD supplementation (expressed in percentages); the orange bar indicates the number of significant taxa in the family that decreased after VD supplementation, over the total number of significant taxa in Firmicutes that decreased after VD supplementation (expressed in percentages).

**Table 1 nutrients-13-03378-t001:** Characteristics of selected studies.

Author, Year	Participants (n°)	Country, Cohort Name	Health status, Inclusion Criteria	Vitamin D Supplementation, Dietary Vitamin D Intakes or 25(OH)D Measure	Microbiota Analysis	Hypervariable Region of 16 sRNA Gene
**Double-blind, randomized controlled trials**						
Ciubotaru, 2015 [27]	115	US	Prediabetes, AAM veteran, aged 35–85 years, BMI 28–39, serum 25(OH) D < 29 ng/mL	ARM1: 400 IU/week + placebo; ARM2: 400 IU/week + 50,000 UI/week for 12 weeks	Ion Torrent Personal Genome Machine	V4
Charoenngam, 2020 [28]	20	US	Healthy adults, serum 25(OH)D levels < 30 ng/mL	Three different arms: 600, 4000 or 10,000 UI/day for 8 weeks	uBiome Inc. (San Francisco, CA, USA).	NR
Hjelmsø, 2020 [29]	580	DK, COPSAC2010 cohort	Pregnant women, gestational age 24 weeks	2800 UI/day from 12 to 16 weeks	Illumina MiSeq	V4
Kanhere, 2018 [30]	38	US	Patients with CF, age ≥ 18 year, no contraindication to oral high-dose vitamin D. Serum 25(OH)D level at baseline 37 ± 6 ng/mL	50,000 UI/week for 12 weeks	Illumina MiSeq	V4
Missailidis, 2019 [31]	23	ET	ART-naïve HIV-positive individuals > 18 years, CD4+ T cells counts > 350 cells/mL, and plasma viral loads > 1000 copies/mL	5000 UI/day (plus phenylbutyrate suppl) for 16 weeks	Illumina MiSeq	V4
Naderpoor, 2018 [32]	26	AU	Healthy adults, serum 25(OH)D levels < 20 ng/mL, BMI > 25, stable weight	100,000 UI at baseline followed by 4000 UI/day for 16 weeks	Illumina MiSeq platform	V6–V8
Sordillo, 2016 [33]	261	US	Pregnant women, aged 18–40 years, gestational age 10–18 weeks	Maternal VDS with 400 or 4000 UI/day for 22–30 weeks	Pyrosequencing 16S RNA gene	V3–V5
**Non-randomized interventional studies**						
Bashir, 2016 [49]	16	AT	Healthy adults, BMI 20–30, non-smokers	980 UI/Kg (week 1–4), 490 UI/Kg (week 5–8)	GS FLX	V1–V2
Bosman, 2019 [51]	21	CA	Healthy adults, aged 19–40 years, Fitzpatrick skin types I–III	Average 1389 UI/day	Illumina MiSeq	V6–V8
Cantarel, 2015 [47]	15	US	Multiple Sclerosis/Healthy women, 25 (OH)D < 30 ng/mL, BMI 18–30	5000 UI/day for 90 days	PhyloChip Array	NR
Garg, 2018 [45]	25	GB	25(OH)D < 50 ng/mL; For UC patients: partial Mayo index of ≤ 4, and stable therapy	40,000 UI/week for 8 weeks	Illumina MiSeq	V3–V4
Schäffler, 2018 [46]	17	DE	CD/Healthy adults, serum 25(OH)D levels < 30 ng/mL	20,000 UI/day 1–3 + 20,000 UI every other day for 4 weeks	Illumina MiSeq	V3–V4
Singh, 2020 [48]	80	QA	Healthy students, serum 25(OH)D levels < 30 ng/mL	50,000 UI/week for 12 weeks	Illumina MiSeq. Metagenomic analysis PICRUST	V3–V4
Tabatabaeizadeh, 2020 [50]	50	IR	Healthy young girls, no history of diabetes, hypertension, or chronic disease	50,000 UI/week for 9 weeks	TaqMan assays	NR
**Observational studies—Cohort**						
Drall, 2020 [34]	1157	CA, CHILD cohort	Pregnant women, gestational age 28 weeks	Maternal and infant VDS of 400 UI/day	Illumina MiSeq platform	V4
Kassem, 2020 [35]	499	US, WHEALS cohort	Pregnant women, aged 21–49 years, gestational ages from 25 to 44 weeks	Maternal serum 25(OH)D and cord blood 25(OH)D levels	Illumina MiSeq	V4
Mandal, 2016 [36]	60	NO, NoMIC cohort	Pregnant women	Dietary VD intakes during 22 weeks of pregnancy: 3.13 µg/day (median)	Illumina MiSeq platform	V4
Talsness, 2017 [37]	913	NL, KOALA cohort	Pregnant women, gestational age 14–18 weeks	Maternal VDS: < or > 400 UI/day for 22–30 weeks. Infant VDS: classified as yes or no	5′- nuclease technique	NR
**Observational studies—Cross-sectional**						
Jackson, 2018 [39]	1724	GB, TwinsUK	Healthy adults	Use of VDS	Illumina MiSeq technology	V4
Luthold, 2017 [40]	150	BR, NutriHS Study	Healthy students, aged 18–40 years, undergraduate or graduate from nutrition colleges	Dietary VD intakes (I: 1.66–4.95/II: 4.97–7.18/III: 7.56–39.87 µg/day)	Illumina MiSeq technology	V4
Seura, 2017 [41]	28	JP	Healthy young women, aged 20–22 years, normal weight	Dietary VD intakes (3.5 ± 2.5 µg/day)	T-RFLP method	NR
Soltys, 2020 [38]	87	SK	UC and CD	Serum 25(OH)D levels	Illumina MiSeq	V4
Thomas, 2020 [42]	567	US	Healthy men (community-dwelling), aged 65 years or older	VDS presents in 424 participants, not quantified. Measure of 25(OH)D; 1,25(OH)2D; 24,25(OH)2D	Illumina bcl2fastq	V4
Wu, 2011 [43]	98	US	Healthy volunteers, aged 2 to 50 years	Dietary VD intakes	454/Roche pyrosequencing. Additional metagenomic analysis with shotgun method	V1–V2
**Observational studies—Case-control**						
Weng, 2019 [44]	113	CN	Age >18 years and confirmed diagnosis of IBD (CD); BMI within the normal range and have not taken any antibiotics, probiotics, prebiotics or yogurt within the previous 4 weeks	Dietary VD intakes	Illumina MiSeq System. Additional metagenomic analysis with shotgun method	V4

25(OH)D—25 hydroxyvitamin D; AAM—African-American men; ART- antiretroviral therapy; BMI—body mass index; CD—Crohn’s disease; FDR—false discovery rate; HIV—human immunodeficiency virus; IBD—inflammatory bowel disease; NR—not reported; PICRUST- phylogenetic investigation of communities by reconstruction of unobserved states; rRNA—ribosomal RNA; SNPs—single nucleotide polymorphism; T-RFLP—terminal restriction fragment length polymorphism; UC—ulcerative colitis; UI—international units; VD—vitamin D; VDR—vitamin D receptor; VDS—vitamin D supplementation.

**Table 2 nutrients-13-03378-t002:** Results of selected studies on alpha and beta diversity with vitamin D supplementation.

Author, Year	Comparison	Serum 25(OH) Levels	Sample	Alpha and Beta Diversity
**Double-blind, randomized controlled trials**				
Ciubotaru, 2015 [27]	Serum 25(OH)D: quintiles	**Baseline:** 14 ± 6 ng/mL **Post:** 36 ± 24 ng/mL	Stool	**Alpha diversity:** NS **Beta diversity:** significant different bacterial composition found in Q1 vs. Q4 of 25(OH)D at genus and family levels
Charoenngam, 2020 [28]	Different doses of VDS	**Baseline:** 16.9 ± 6.0 ng/mL; 20.3 ± 6.3 ng/mL; 18.5 ± 3.5 ng/mL **Post:** 20.0 ± 3.4 ng/mL; 39.0 ± 8.7 ng/mL; 67.3 ± 3.1 ng/mL	Stool	**Alpha diversity:** NS **Beta diversity:** NR
Hjelmsø, 2020 [29]	Different doses of prenatal VDS	Not reported	Infant stool	**Alpha diversity:** NS **Beta diversity:** NS
Kanhere, 2018 [30]	Supplemented group vs placebo group in vit D insufficient at baseline	**Baseline:** VD suff: 37 ± 6 ng/mL; VD insuff, Pl.: 22 ± 6; VD insuff, suppl.: 25 ± 5 ng/mL **Post:** VD insuff, Pl.: 25 ng/mL; VD insuff, suppl.: 45 ng/mL	Stool	**Alpha diversity:** NS **Beta diversity:** significantly different composition at follow-up in the supplemented group compared to the placebo
Missailidis, 2019 [31]	Supplemented group versus placebo group	**Baseline:** NR **Post:** NR	Mucosal gut biopsy	**Alpha diversity:** NS **Beta diversity:** NS
Naderpoor, 2019 [32]	Supplemented group versus placebo group	**Baseline:** VD group 31.54 ± 4.4 vs. Pl 31.07 ± 4.1 nmol/L **Post:** VD group 91.14 ± 25.8 vs. Pl 31.58 ± 14.11 nmol/L	Stool	**Alpha diversity:** significant reduction in richness at follow-up in the supplemented group **Beta diversity:** significant difference in composition between groups at follow-up at the genus level
Sordillo, 2016 [33]	Maternal VDS Umbilical cordon 25(OH)D levels	**Baseline:** 22.7 ± 11.9 ng/mL	Stool	**Alpha diversity:** NS **Beta diversity:** NR
**Non-randomized interventional trials**				
Bashir, 2016 [49]	Post- versus pre-supplementation	**Baseline:** 22.3 ± 13.1 ng/mL **Post:** 55.2 ± 13.3 ng/mL	Biopsy and stool	**Alpha diversity:** significant increased richness in GA **Beta diversity:** significant change in composition only in upper GI tract
Bosman, 2019 [51]	Prior VD supplemented group (VDS+) vs prior non-supplemented group (VDS-) before UVB exposure	**Baseline:** NR **Post:** NR	Stool	**Alpha diversity:** VDS- showed significantly lower diversity and richness before UVB exposure than VDS+ **Beta diversity:** NR
Cantarel, 2015 [47]	Post- versus pre-supplementation in healthy controls and in patients with multiple sclerosis	**Baseline:** 23.2 ± 5.7 ng/mL in the HCs; 25.9 ± 4.4 ng/mL in MS **Post:** 59.8 ± 11.7 ng/mL in the HCs; 55.6 ± 17.0 ng/mL in MSs	Stool	**Alpha diversity:**NS **Beta diversity:** NS
Garg, 2018 [45]	Post versus pre-supplementation	**Baseline:** 34 (range 12–49) nmol/L **Post:** 111 (range 71–158) nmol/L	Stool	**Alpha diversity:** NS **Beta diversity:** NS
Schäffler, 2018 [46]	Post versus pre-supplementation in healthy controls and in patients with CD	**Baseline:** in CD 39.7 ± 23 nmol/L, in HC 29.6 ± 6.3 nmol/L **Post:** in CD 121.4 ± 43.2 nmol/L, in HC 143.0 ± 25.2 nmol/L	Stool	**Alpha diversity:** In HC, NS; in CD taxa significantly decreased after VDS. **Beta diversity:** NS
Singh, 2020 [48]	Post- versus pre-supplementation	**Baseline:** 11.03 ± 0.51 ng/mL **Post:** 34.37 ± 1.47 ng/mL	Stool	**Alpha diversity:** Significant increase in observed OTUs and Chao1 indices, no difference in Shannon Index. **Beta diversity:** significant difference in composition between vs pro supplementation
Tabatabaeizadeh, 2020 [50]	Post- versus pre-supplementation	**Baseline:** 11 ± 9 ng/mL **Post:** 40 ± 17 ng/mL	Stool	**Alpha diversity:** NR **Beta diversity:** NR
**Observational studies—Cohort**				
Drall, 2020 [34]	Pre vs post maternal VDS and Infant VDS	**Baseline:** NR **Post:** NR	Stool	**Alpha diversity:** NR **Beta diversity:** NR
Talsness, 2017 [37]	Comparisons of 3 levels of maternal VDS; Infant VDS vs non-infant VDS	**Baseline:** 44.3 ± 18.3 nmol/L **Post:** NR	Stool	NR

25(OH)D—25 hydroxyvitamin D; CD—Crohn’s disease; FDR—false discovery rate; HC—healthy controls; ND-UVB—narrow-band ultraviolet-B; NR—not reported; NS –not significant; SNPs—single nucleotide polymorphisms; Pl—placebo; UC—ulcerative colitis; VD—vitamin D; VDR—vitamin D receptor; VDS—vitamin D supplementation; OTUs: operational taxonomic units.

**Table 3 nutrients-13-03378-t003:** Results of selected studies on alpha and beta diversity with Vitamin D Intake or Serum Concentrations.

Author, Year	Comparison	Serum 25(OH) Levels	Sample	Alfa and Beta Diversity
**Observational studies—Cohort**				
Kassem, 2020 [35]	Maternal serum 25(OH)D levels Umbilical cord blood 25(OH)D levels	**Baseline maternal serum 25(OH)D**: 25.04 ± 11.62 ng/mL **Baseline umbilical cord blood 25(OH)D levels**: 10.88 ± 6.77 ng/mL	Stool	**Alpha diversity**: Prenatal 25(OH)D level significantly associated with decreased infant richness and diversity at 1 month; cord 25(OH)D level was positively associated with infant gut evenness in White women and negatively associated with infant evenness at 6 months. **Beta diversity:** both prenatal and cord 25(OH)D were significantly associated with 1-month composition
Mandal, 2016 [36]	Dietary maternal VD intakes	**Baseline:** Not reported **Post:** Not reported		**Alpha diversity:** VD intake was significantly and inversely associated to whole tree phylogenetic and Shannon diversity **Beta diversity:** NS
**Observational studies—Cross-sectional**				
Jackson, 2018 [39]	Intake of VD supplements: yes versus no	NR	Stool	**Alpha diversity:** NS **Beta diversity:** NS
Luthold, 2017 [40]	Dietary VD intakes: high versus low-tertile Serum 25(OH)D leves: high- versus low tertile	**Baseline:** 23.9 ± 9.7 ng/mL	Stool	**Alpha diversity:** NR **Beta diversity:** NR
Seura, 2017 [41]	Dietary VD intakes	**Baseline:** NR	Stool	**Alpha diversity:** NR **Beta diversity:** NR
Soltys, 2020 [38]	Serum VD levels in patients with UC and CD	**Baseline:** in winter/spring 25.05 ng/mL and summer/autumn period 37.26 ng/mL	Biopsy and stool	**Alpha diversity:** NR **Beta diversity:** NS
Thomas, 2020 [42]	VD metabolites	25(OH)D (34.2 ng/mL), 1,25(OH)2D (56 pg/mL), and 24,25(OH)2D (3.2 ng/mL)	Stool	**Alpha diversity:** 1,25(OH)2D, active ratio and catabolism ratio are positively and significantly associated with diversity. **Beta diversity:** 1,25(OH)_2_D, 24,25(OH)_2_D, activation ratio, catabolism ratio significantly define clusters of microbial composition
Wu, 2011 [43]	Dietary VD intakes	**Baseline:** NR	Stool	**Alpha diversity:** NR **Beta diversity:** NR
**Observational studies—Case-control**				
Weng, 2019 [44]	Dietary VD intakes in healthy controls and patients with UC and CD	**Baseline:** NR	Biopsy and stool	**Alpha diversity:** NR in UC **Beta diversity:** NR

25(OH)D*—*25 hydroxyvitamin D; CD*—*Crohn’s disease; FDR*—*false discovery rate; HC*—*healthy controls; ND-UVB*—*narrow-band ultraviolet-B; NR*—*not reported; NS—not significant; SNPs*—*single nucleotide polymorphisms; Pl—placebo; UC*—*ulcerative colitis; VD*—*vitamin D; VDR*—*vitamin D receptor; VDS*—*vitamin D supplementation; OTUs: operational taxonomic units.

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
