# Peer review of "The Association between Vitamin D and Gut Microbiota: A Systematic Review of Human Studies"

_nutrients, 2021, doi:10.3390/nu13103378_

Round 1

Reviewer 1 Report

In the manuscript “The association between vitamin D and gut microbiota: A systematic review of human studies” by Bellerba et al., the authors have done a systematic review of the literature on the effect of vitamin D intake on the gut microbiota.

25 studies are included in their qualitative analysis, with the main focus of covering if supplementation or serum concentration is associated with increase or decrease of specific bacterial species, beta diversity changes, and alpha diversity. 

They find that individual studies vary a lot, but find that supplementation is associated with changes in the bacterial phyla Firmicutes, Actinobacteria and Bacteroidetes (Which is also the most common in the gut). 

Whilst the Introduction and literature search strategy seems to be fine, I find that the review part of the manuscript seems lacking. Specifically, the lack of any kind of quantitative formal meta analysis seems unhelpful, as the readers of the manuscript are only presented with an index of previously published studies. 

The figures of the manuscript does not convey any helpful information, and does not help in unravelling the difference between spurious results and more robust findings when evaluating all 25 studies. 

Similarly the phylogenetic reconstruction in the supplementary table S1-S4 is also of limited use, and does not add anything of value to the readers. 

Finally I think the manuscript would benefit from a read through from a person with English as their first language, as I sometimes struggled to understand the meaning of specific sentences. 

In conclusion, I think the authors need to add more analysis to pinpoint the most robust associations between Vitamin D and the gut microbiota. This would be helpful to other researchers in this field, and would certainly be worthy of publication. 

Author Response

Thank you for the suggestions. We added further analyses to summarize results and we presented them with different figures in order to make conclusions clearer.

Reviewer 2 Report

I read with interest the article submitted by Bellerba et al and colleagues in which they systematically reviewed the human evidence on vitamin D and gut microbiota. I think this is a timely topic. Despite the increasing studies reporting associations between vitamin D intake/supplementation or circulating vitamin D levels and gut microbiota composition, there has been a lack of comprehensive summary of evidence from human studies. Nevertheless, I still have several comments:

  1. In Introduction, the authors stated that “VD deficiency affects the expression of VDR and subsequently the integrity of intestinal barrier and microbial composition.” Doe VD deficiency directly affects VDR expression in the gut? I think this is an important statement supporting a real/direct effect of vitamin D on gut microbiota. It is helpful if the authors could expand this and add the reference.
  2. Related to above, in Discussion, the authors discussed potential mechanisms underlying the effects of VD on gut microbiota composition. It is unclear why “these results must be interpreted with caution”. Please explain.
  3. There is inconsistency regarding the comment on studies using shotgun metagenomic sequencing. It seems none of the studies included have used shotgun metagenomic sequencing?

Results, “Moreover, only three studies conducted metagenomic sequencing by shotgun[34,39,40].

Discussion, “In contrast, a genome-wide shotgun sequencing approach ensures a better taxonomic analysis and genomic information, but this method was used only in two of the selected studies for this review[40,44].

  1. The conclusion is inaccurate. Well-designed RCTs could minimize the biases of the study (confounding, reverse causation, etc), but more information will be needed to infer the mechanisms, such as functional changes in response to VD supplementation/status with functional profiling of the gut microbiome or stool metabolomics. Evidence from animal studies, coupled with human evidence, may also aid the understanding of the mechanisms.
  2. From the figures of phylogenic tree of taxa that were significantly changed after vitamin D supplementation, a notable finding is the highly inconsistent results across studies. This should be further highlighted.

Author Response

We thank the editor and reviewers for their helpful suggestions. Hereafter please find a point-by-point reply to reviewers’ comments. We highlighted in the article the main changes made from the previous paper .

  1. In Introduction, the authors stated that “VD deficiency affects the expression of VDR and subsequently the integrity of intestinal barrier and microbial composition.” Doe VD deficiency directly affects VDR expression in the gut? I think this is an important statement supporting a real/direct effect of vitamin D on gut microbiota. It is helpful if the authors could expand this and add the reference.

We agree with the Referee’s comment and we added the following paragraph in the introduction about this aspect:

“However, little is known about the direct effects of vitamin D on bacteria. VD inhibits the growth of specific mycobacterial species in vitro (Greenstein RJ, et al 2012) implicating that microbial effects of vitamin D would be consistent. With known immunoregulatory proprieties.. Jahani et al (Jahani, R., et 2014,) shows that in mice exposed to high vitamin D3 during pregnancy and lactation, lower level of vitamin D resulted in reduced vitamin D receptor and increased expression of pro-inflammatory genes in the colon at 3 months, lower numbers of colonic Bacteroides/Prevotella at postnatal day 21 and higher serum LPS concentration at adulthood. These results are confirmed by other rodent studies that demonstrate that vitamin D deficiency by dietary restriction, lack of CYP27B1, or lack of VDR promote increases in the Bacteriodetes (Ooi eta, 2013; Assa et al, 2014; Jin et al, 2015; Wu et al, 2015) and Proteobacteria phyla (Ooi et al 2013; Assa et al, 2014; Wu et al 2015). Furthermore, a GWAS carried out in a combined cohort of 2029 individuals identified two VDR polymorphisms as significant contributors to microbiota variation (Wang J, et al 2016). The human VDR polymorphisms  consistently influenced the genus Parabacterioides (phylum: Bacterioidetes), and subsequent evaluation of VDR−/− mice showed a corresponding increased abundance of Parabacteroides compared to WT mice (Wang J, et al 2016).”

  • Greenstein, R. J., Su, L., & Brown, S. T. (2012). Vitamins A & D inhibit the growth of mycobacteria in radiometric culture. PloS One, 7(1), e29631. https://doi.org/10.1371/journal.pone.0029631
  • Jahani, R., Fielding, K. A., Chen, J., Villa, C. R., Castelli, L. M., Ward, W. E., & Comelli, E. M. (2014). Low vitamin D status throughout life results in an inflammatory prone status but does not alter bone mineral or strength in healthy 3-month-old CD-1 male mice. Molecular Nutrition & Food Research, 58(7), 1491–1501. https://doi.org/10.1002/mnfr.201300928
  • Ooi JH, Li Y, Rogers CJ, Cantorna MT. Vitamin D regulates the gut microbiome and protects mice from dextran sodium sulfate – induced colitis 1 – 3. J Nutr. (2013) 143:1679–86. doi: 10.3945/jn.113.180794
  • Assa A, Vong L, Pinnell LJ, Avitzur N, Johnson-henry KC, Sherman PM. Vitamin D deficiency promotes epithelial barrier dysfunction and intestinal inflammation. J Infect Dis. (2014) 210:1296–305. doi: 10.1093/infdis/jiu235
  • Jin D, Wu S, Zhang Y, Lu R, Xia Y, Dong H, et al. Lack of vitamin D receptor causes dysbiosis and changes the functions of the murine intestinal microbiome. Clin Ther. (2015) 37:996–1009.e7. doi: 10.1016/j.clinthera.2015.04.004
  • Wu S, Zhang YG, Lu R, Xia Y, Zhou D, Petrof EO, et al. Intestinal epithelial vitamin D receptor deletion leads to defective autophagy in colitis. Gut. (2015) 64:1082–94. doi: 10.1136/gutjnl-2014-307436

  1. Related to above, in Discussion, the authors discussed potential mechanisms underlying the effects of VD on gut microbiota composition. It is unclear why “these results must be interpreted with caution”. Please explain.

Thank you for pointing this out. In fact the comment did not refer to the previous sentence, thus we rephrased the sentence.

  1. There is inconsistency regarding the comment on studies using shotgun metagenomic sequencing. It seems none of the studies included have used shotgun metagenomic sequencing?

We checked again and 2 studies (Wu e Weng) carried out also shotgun analysis in a subgrup of patients. The sentence and table 1 have been corrected accordingly.

  1. The conclusion is inaccurate. Well-designed RCTs could minimize the biases of the study (confounding, reverse causation, etc), but more information will be needed to infer the mechanisms, such as functional changes in response to VD supplementation/status with functional profiling of the gut microbiome or stool metabolomics. Evidence from animal studies, coupled with human evidence, may also aid the understanding of the mechanisms.

Mouse models studies help to better understand the functional mechanisms, for example by using, in certain pathological contexts, VD supplementation in concomitance with specific systems to evaluate the function and composition of the microbiota (e.g. using GF mice, or treated with selective antibiotics, or using stool transplantation from patients treated or not with VitD). For example, in the study by Bora et al. they investigated the effect of microbiota on vitamin D metabolism in germ free mice, and found several factors responsible for the modulation of this relationship, such as FGF23 which decreased vitamin D, highlighting the importance of investigating the mechanisms of this process (Stephanie A. Bora, Mary J. Kennett, Philip B. Smith, Andrew D. Patterson, and Margherita T. Cantorna. Front Immunol. 2018; 9: 408. Published online 2018 Mar 2. doi: 10.3389/fimmu.2018.00408 The Gut Microbiota Regulates Endocrine Vitamin D Metabo-lism through Fibroblast Growth Factor.)

  1. From the figures of phylogenic tree of taxa that were significantly changed after vitamin D supplementation, a notable finding is the highly inconsistent results across studies. This should be further highlighted.

Thank you for the suggestions. We added a sentence about this aspect in the discussion

Round 2

Reviewer 1 Report

  • Figur 1 is not a mirror barchart. And I would like a description of the colors (red /blue) in the legend.
  • Figure 2. I think the legend text is a little hard to understand. So for example in 2.A: Lacnospiracaea was decreased in 24% of the studies and increased in 17% of the studies in the RCT group that received Vitamin D. ? Or am I understanding it wrong?

I still think the phylogenic trees (figure 3, 4 and 5) add very little to none, to the readers and should be removed/ moved to the supplementary analysis. To me your manuscript is a fine summary of the current litterature, and the tables and figure 1 and 2 are enough.

I do like figure 2/the mirrored barchart. However I think the results of that figure should be more emphasized in the paper: That there seems to be very conflicting result in the current litterature.